# Thoracoscopic Guided Pericostal Sutures as a Solid Fixation for Primary Closure of Congenital Diaphragmatic Hernias

**DOI:** 10.3390/children9081116

**Published:** 2022-07-26

**Authors:** Armin-Johannes Michel, Ulrike Metzger, Steven Alan Rice, Roman Metzger

**Affiliations:** 1Department of Pediatric and Adolescent Surgery, Paracelsus Medical University Hospital, 5020 Salzburg, Austria; u.metzger@salk.at (U.M.); r.metzger@salk.at (R.M.); 2University Hospital for Children and Adolescents, Universitätsmedizin Rostock, 18057 Rostock, Germany; steven.rice@uni-rostock.de

**Keywords:** repair of congenital diaphragmatic hernia, CDH, recurrence of CDH, thoracoscopic pericostal suture technique

## Abstract

**Purpose:** To describe a minimally invasive technique with primary closure and strong suture connection that is feasible in cases of larger, most common type B defects of congenital diaphragmatic hernia (CDH). **Background:** The thoracoscopic approach (TA) is a favorable technique for the repair of CDH and is still evolving globally. A common issue is finding the optimal suture technique for secure closure in order to prevent recurrences. Whether a defect can be closed only by sutures or by using a patch depends on the size of CDH, the presence of a muscular rim along the inner thoracic surface and finally on the surgeon’s experience. From a geometrical point of view, the challenge is to transform the circular defect into a line, without tension, with a strong compound and preferably without additional material. To address this, we apply a setting of the sutures in a “T-shape” and a way to lead the sutures around the rib bones in order to increase stability. This method allows for the primary closure of CDHs and also applies to larger defects. **Cases:** We present seven newborns with posterolateral CDH on the left side. The defects were solely repaired by TA and by the suturing technique described in detail.

## 1. Introduction

CDH is a rare anomaly usually detected prenatally by ultrasound. Magnetic resonance imaging (MRI) is used additionally for complex tasks. In 90% of the patients, the defect occurs in the posterolateral area of the diaphragm, and in over 85% of patients, it occurs on the left side [1]. The results of the European Surveillance of Congenital Anomalies (EUROCAT) show a total prevalence rate of 2.30 (1.39 to 4.10) per 10,000 births in Europe with a small increase over the years [2]. The displacement of abdominal organs into the thoracic cavity can cause deficient pulmonary development. A hypoplastic lung could necessitate in rare cases extracorporeal membrane oxygenation (ECMO) to ensure survival. In general, neonates with CDH require intensive care and delicate adjustment of the cardiorespiratory system with emphasis on pulmonary pressure and right-to-left shunt. Overall survival has improved in the past decades, but the mortality for isolated CDH is still remaining at 25 to 30 percent. This is strongly connected to the size of the defect and to major cardiac anomalies [3,4]. After stabilization of the cardiorespiratory system is achieved by the neonatologists and anesthesiologists, the defect needs to be closed without delay. It is for the surgeon to decide between open (OA) or TA and, mostly intraoperative, whether there is a prosthetic patch needed.

Since 2014 we have repaired CDH using TA. In order to adapt the muscle edges, we connect them with the inner surface of the thoracic wall by single sutures. Larger defects used to be closed with a prosthetic patch (VYPRO^TM^ II, Vicryl-Prolene-composite, Ethicon, Inc., Route 22 West, Post Office Box 151, Somerville, NJ 08876, USA). However, a patch is a foreign object which can lead to worse functioning of the diaphragm because of its inability to adapt to the body growth of the patient. It is seen as a predisposing factor for the recurrence of CDH [5]. Despite this concern, we had no recurrent CDH after TA, we wanted to close the defect more physiologically and avoid patches. Therefore, we developed a special suture pattern with a pericostal suture placement in 2016. 

## 2. Methods

A retrospective analysis was performed reviewing the medical data between 2016 and 2021 of seven neonates with a thoracoscopic repaired, left-sided CDH.

**Preoperative Setting:** After delivery, a plane X-ray image is taken at the intensive care unit (ICU) (see Figure 1). To rule out other associated anomalies, echocardiography and ultrasound of the brain and the abdominal organs are necessary. All patients were intubated between the first 2–4 h after birth in general to prevent abdominal distension and to avoid an additional compromise of the lung function. As soon as the patient is in a stable cardiorespiratory condition the surgical repair of the defect is performed. The patient is placed on the OR table in lateral decubitus position, the side of the defect upwards with an axillary roll underneath it, and the body is appropriately padded. The preoperative antiseptic routine is performed. Under complete relaxation by anesthesia three 3-mm trocars are placed in the standard location for CDH repair: one at the lower tip of the scapula for the camera, one in the frontal axillary line and one in the posterior. Insufflation pressures during operations are 3–6 mmHg with a flow of 1 L/min. Single lung ventilation is not needed. The herniated bowel is placed downwards into the abdomen and the size and nature of the defect are evaluated.

**Intraoperative situation:** Once the defect is visible thoracoscopically, the center of the opposing diaphragmatic margins is taken by two graspers and pulled in a T-shape toward the thoracic wall. This is to verify the tension and the possibility of closing the gap using primary sutures. The margin of the pleuro-peritoneal canal is incised circumferentially to create an iatrogenic wound to the adjacent surfaces. This is considered beneficial for subsequent scarring and to improve the strength of the compound (see Figure 2a,b). Outside the thorax, at the expected central point of suture (CPS) we set an incision of 3 mm into the skin. At each center of the opposing diaphragmatic edges 3.0 Ethibond threads (Ethicon, Inc., Route 22 West, Post Office Box 151, Somerville, NJ 08876, USA) are placed (see Figure 3a,b). The sutures are laid, one superior, one inferior, around the rib (see Figure 4). The needles of the corresponding sutures are pierced from intrathoracic towards extrathoracic. The subcostal needle must be guided in contact with the bone’s circumference in order to avoid potential injury to the subcostal nerve and vessels. Now, the threads pass the thoracic wall and the skin incision. The knots are pulled down slowly by hand from the outside to control the traction (see Figure 3a). The camera observes the approximating edges until the gap is closed. This maneuver is the core element for the successful closure of the gap only using sutures. To complete the fixation of the defect and as a reinforcement, up to seven additional sutures are positioned in the same pericostal way. It may also be necessary to include adjacent ribs through pericostal sutures. Further, single sutures are placed to close the remaining muscular defect. Finally, the edges move towards each other in a T-shaped figure (see Figure 5a,b). Subcutaneous sutures are not necessary. All skin incisions are sealed by glue. A pleural drainage Ch 18 without suction is placed through the ventral access of the trocar.

A control X-ray is taken on the ICU 2–4 h after the end of the operation (see Figure 6). All operations were conducted by the same surgeon.

## 3. Results

**Patient Data:** There were six male, and one female, patients with left-sided CDH born between 2016 and 2021. Three patients had no history of prenatal diagnosis of CDH because of non-compliance with prenatal screening. They were delivered spontaneously but conspicuously due to reduced breathing with the unilateral excursion of the thorax. As the diagnosis was proved by plane X-ray, intubation was performed in the first 4 h after birth. Four patients were found to have CDH during prenatal ultrasound. The mean lung head ratio (LHR) was 2.71 (range 1.8–3.49) and they were delivered by cesarian section. Of all seven patients, the mean age at delivery was 38/4 weeks (+/−14 days). APGAR scores at 1 min after birth were 7.2 (5–9), after 5 min 7.6 (6–10) and 10 min 8.5 (7–10). One baby was born in an emergency car and APGAR scoring was not possible. The time from delivery to intubation was 71 min on average (20–270 min). Median birth weight was 3190 g (+/−1010 g). Median time from birth to operation was 19 h (+/−11.5 h).

**Intraoperative findings:** All seven patients had a left-sided CDH in Bochdalek’s position with a size near 50% of the size of the hemilateral diaphragm. The reduction of the herniated viscera proceeded without complications. An abnormal enhancement of the CO_2_ pressure was not observed. All CDHs were closed by primary suture. An additional resection of a Meckel’s diverticulum was performed intraoperatively in one patient and in another patient, we resected an extralobar sequester of the lung by a stapler maneuver. The average duration time for thoracoscopic repair of the CDH was 158.8 min. (range 104–295 min). No complications affecting the cardiorespiratory system of the newborns occurred during or after all seven operations.

**Postoperative follow-up:** All patients left our operation room intubated and in cardiorespiratory stable condition. The mean ventilation time after operation including weaning was 75.8 h (range 25.5–132 h). The mean length of hospital stay was 19 days (range of 7–33 days). Postoperative clinical follow-ups were performed 2 and 6 months after hospital discharge with plane X-ray images (see Figure 7). Further follow-up examinations were performed every six months up to two years of age, thereupon, they were performed every year. Today, the mean follow-up time is 3.5 (1.1–6.3) years but in the future, the number will grow. Each patient was seen on average 5 (3–9) times after discharge and they remain in our follow-up schedule. The outcome of all seven patients to date was excellent, we had no recurrent hernias, no scoliosis, and no cases of intercostal neuropathy.

## 4. Discussion

The presented seven patients were born between 2016 and 2021. After delivery, none of them displayed complications affecting the cardiorespiratory system, further severe anomalies were ruled out. The decision to correct the defect by the method presented here was based on these specific criteria. Between 2016 and 2021 we had four additional patients with CDH. They could not be included in this study because it was not possible to perform the technique; one patient had a muscular defect on the right side with liver elevation. The defect was closed by TA without pericostal sutures. The other three patients were closed by OA; two patients had a complete left-sided CDH and needed a large patch. One patient, also left-sided, had additional malrotation of the intestine with ectopic pancreatic tissue.

What is a small defect of CDH, what constitutes a large one, and where is the limit for primary closure?

The Congenital Diaphragmatic Hernia Study Group (CDHSG) recommended in 2013 a differentiation of CDH into four types (A–D) based on the size of the defect. Defect A is described as a gap, which is surrounded by the unilateral diaphragmatic muscle. Defect B is adjacent to the inner thoracic wall and the area is smaller than 50% of the hemithoracic part of the diaphragm. C, on the other hand, is larger than 50%. D affects the whole unilateral diaphragm comparable to an aplasia [6]. Taking this classification into account since 2016 we used to close larger B graded defects by TA with primary suture which are near to 50% of the hemithoracic muscle or on the border to type C. We modified the technique by involving the rib bone in the suture. Thereby a counterpoise for the stitches going through the muscle is provided because the inner surface of the thoracic wall is mechanically a weak point. Compared to the method we performed before 2016, in large B graded defects this technique made patches obsolete.

In general, TA leads to a higher rate of recurrences compared to OA [7] but patients need less ventilatory support after minimally invasive surgery [8]. The results of the CDHSG showed that a defect type B (≤50% of the unilateral diaphragm area) occurs in 716 of 1638 patients (43.7%). This is the most common size of CDH compared to the other types: A was seen in 13.3%, C in 30.2% and D in 12.8% of the cases. Concerning the side of the defect, it is type B which is also seen mostly on the left side (87.2%) [6]. According to this data, the technique we present gives a solution for the management of the most common variant of CDH. In the context of the declining use of prosthetic patches, which are known for greater risks of chylothorax, recurrences and small bowel obstruction [9], our method offers distinct advantages. Limits are determined by the thickness and the elasticity of the diaphragmatic muscle around the defect. An intraoperative maneuver to test diaphragmic tension, monitoring of the patient’s vital signs, and surgical experience are required to determine whether the primary closure would work.

Previous suture techniques in TA are described: Slipknot-tying [10] with or without using a patch as well as nonabsorbable silk 2.0 sutures guided through a 16 G injection needle [11]. Both reports do not describe a pericostal fixation. In one case report, pericostal suturing is featured. The intrathoracic suture is guided outside by a Tuohy needle [12]. A Tuohy needle is a cannula that is used for peridural anesthesia. The author describes the way the inserted cannula picks up the needle of the intrathoracic suture and guides it outwards around the rib. It may serve as an unerringly support for the placement of the subcutaneous end of the pericostal sutures. This method seems very effective for saving time. The edge of the diaphragmatic muscle is pulled directly towards the thoracic wall, so the closure of the circular defect ends up being C-shaped. The tension of these sutures is supposed to be stronger than that of the T-shaped we mentioned.

A noteworthy report was published in 2020 by Bogusz et al. [13]. They describe an adapted method of percutaneous internal ring suturing (PIRS). This method is derived from the repair of pediatric inguinal hernias by introducing the sutures through an injection needle. In this report, the pericostal suturing technique is quite similar to ours. The probable impairment of the subcostal nerve and vessels is not considered. In addition, the difference is that the sutures are inserted through a loop that is pulled out of the chest using an 18 G needle. We performed percutaneous insertion of the suture directly using the original needle of the 3.0 Ethibond suture, which corresponds better with the anatomical conditions—the placement of the diaphragmatic edges and the protection of the subcostal structures. The scarification of the diaphragmatic margins and the corresponding rib is also described. However, it is not mentioned whether it is a sharp wound, or if it happened through coagulation. In our cases, we cut the surface with a scalpel. The results were small hemorrhages.

The technique presented, including the iatrogenic lesions of the diaphragmatic margins and the parietal pleura, the T-shaped placement focusing on a CPS and the pericostal, subcutaneous placement through only one skin incision, are the essential contents of this report. It is meant as a contribution to help increase the overall success of the interdisciplinary recovery process in neonates with CDH. At our hospital, all patients with CDH will be followed up with every year until adulthood. In conclusion, this method, performed with TA, has so far provided a normal quality of life and recurrence-free CDH repair from 2016 to date.

## Figures and Tables

**Figure 1 children-09-01116-f001:**
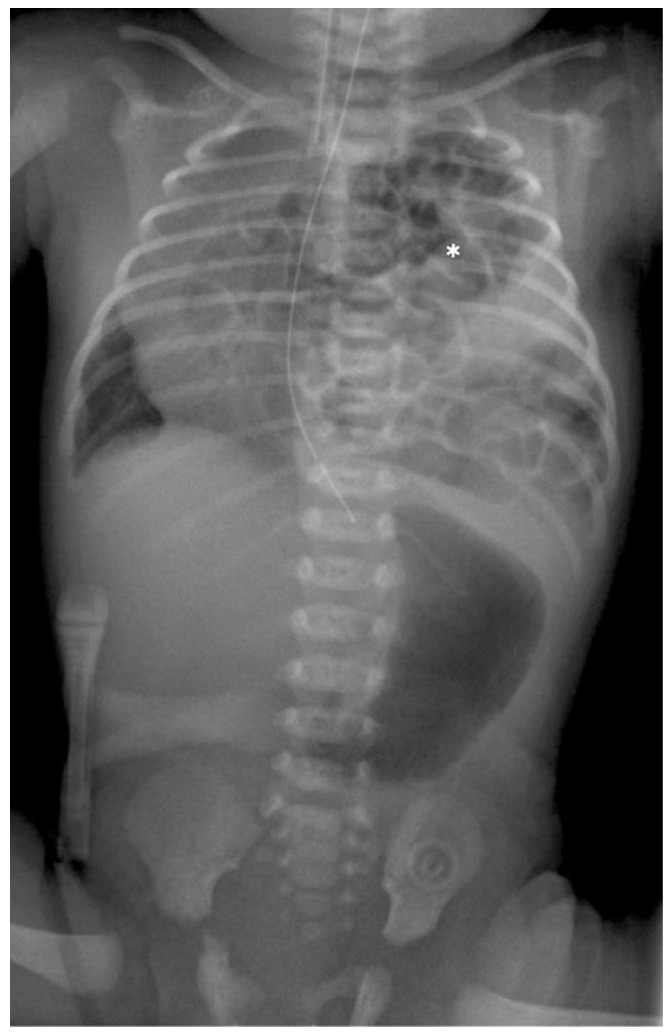
Preoperative radiograph of a left-sided CDH. The * indicates the intrathoracic bowel.

**Figure 2 children-09-01116-f002:**
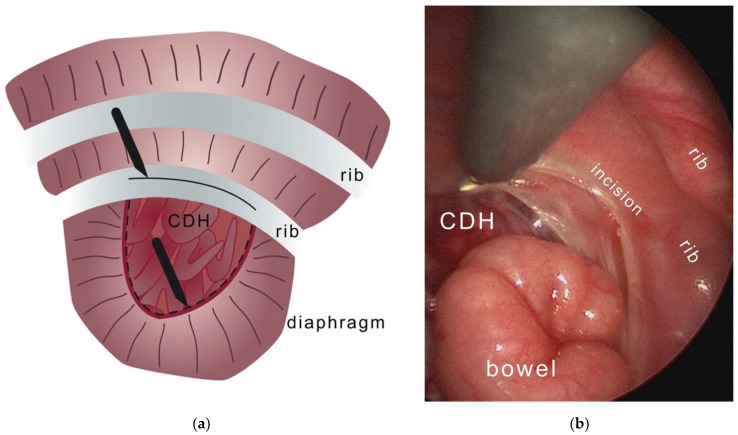
Incision of the pleuroperitoneal canal along the rib and at the edge of the diaphragmatic border. (**a**) Illustration; (**b**) Photograph of the intraoperative situs.

**Figure 3 children-09-01116-f003:**
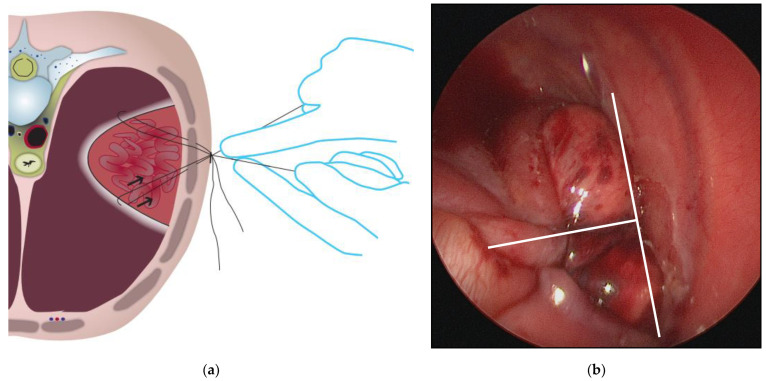
The placement of the sutures inside the thorax and the prove of tension from outside. (**a**) Illustration; (**b**) Photograph of the intraoperative situs with the expected T-shaped line of the suture.

**Figure 4 children-09-01116-f004:**
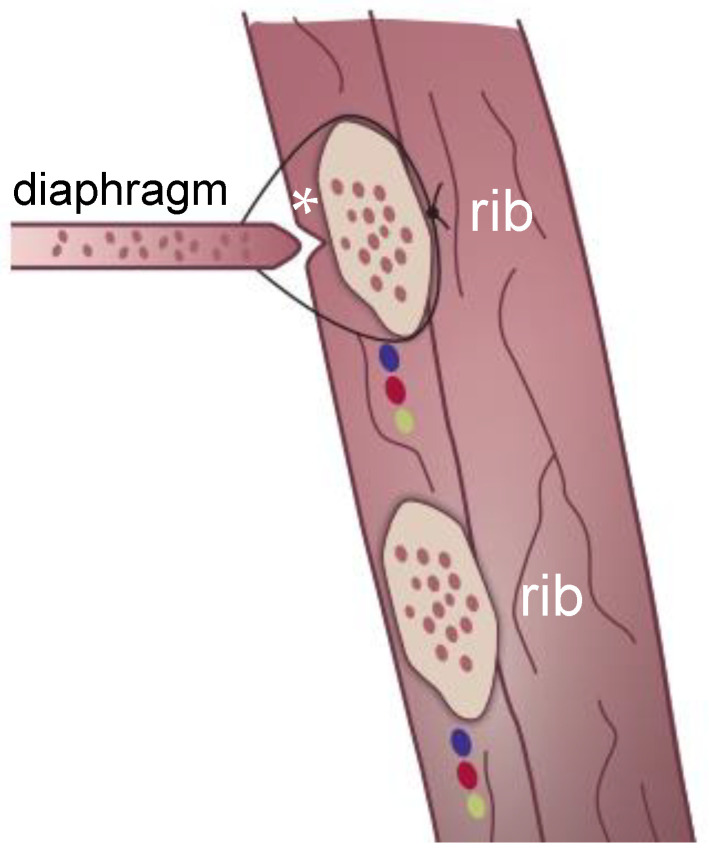
The seam runs around the rib, preferably located between the rib and the subcostal vessels. Further along are the cut (*) in the pleuroperitoneal channel and the ideal insertion of the diaphragm at the rib.

**Figure 5 children-09-01116-f005:**
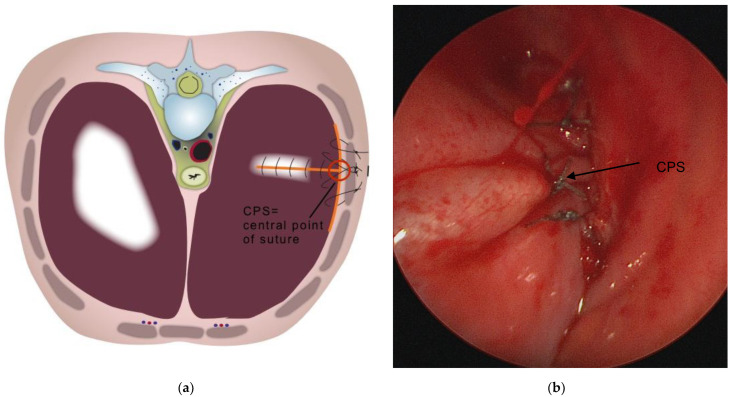
The final CDH T-shape repair and the placement of the sutures. (**a**) Illustration; (**b**) Photograph of the intraoperative situs.

**Figure 6 children-09-01116-f006:**
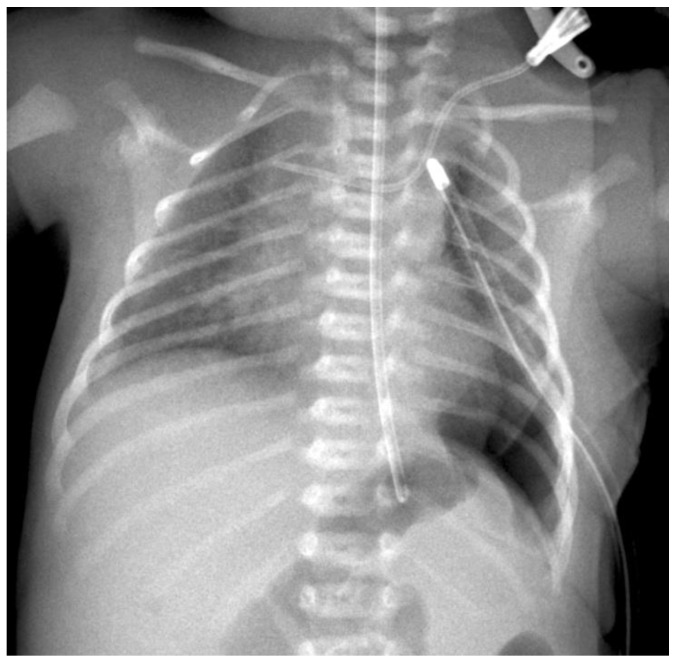
The postoperative radiograph.

**Figure 7 children-09-01116-f007:**
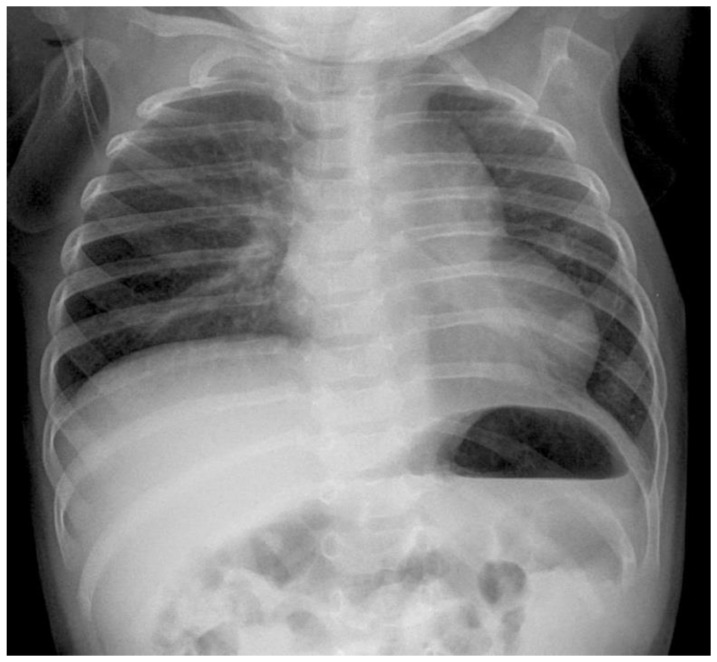
Radiograph of the same patient after 6 months.

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
