# Peer review of "Thoracoscopic Guided Pericostal Sutures as a Solid Fixation for Primary Closure of Congenital Diaphragmatic Hernias"

_children, 2022, doi:10.3390/children9081116_

Round 1
Reviewer 1 Report
Comments from the reviewer:
The authors described in this paper a new technique with thoracoscopic approach for the congenital diaphragmatic repair. This minimal invasive technique is well described and well-illustrated. The relevance and the interest of the paper would increase if the authors specified the context of patients that would benefit from this technique.
Introduction section:
This paragraph should be improved. The authors have to precise the aim of the paper: to develop new approach of closing the diaphragmatic defect without patch that could be used for large defects or to describe a new different thoracoscopic technique for the diaphragmatic repair compared to the previous published thoracoscopic approaches.
The new approach of the authors relies on previous thoracoscopic technique (Boguzs et al.) or others one that could be more detailed in this paragraph to understand why the authors decided to “change” the way to setting the seams.
Patients and methods section:
Patients and methods section should be rewritten, especially the methods of inclusion of patients for the choice of the surgical technique, details of the data collected for each patient.
Patients’ data (Lines 38 to 43) should be included in result section, these data represent descriptive results.
L38. Were the seven patients successive patients on the study period? How many CDH patients are being treated each year in their hospital and with what surgical technique ?
L39. What were the circumstances and delay of the diagnosis of CDH of the patients without prenatal diagnosis? Were these patients intubated before the surgical repair or only for surgical repair?
L40. It would be interesting to know the antenatal characteristics for antenatally diagnosed patients (LHR o/e; pulmonary volume o/e; liver position..)
L41. Could the authors explain the delivery by cesarian section for the patients with prenatal diagnosis? Were these patients intubated at birth or only for the surgical repair? What are the medical conditions to perform thoracoscopic repair (including ventilation and pulmonary hypertension) ? Is thoracoscopic repair feasible as a bedside surgery in their center ?
Methods
Intraoperative situation: Why did the authors use 3.0 Ethibond and not 2.0 Ethibond ?
Results section:
This very short section should be completed according to these following points:
Intraoperative findings:
- mean operative duration?
- size of the defect? The authors mentioned in methods section that “the size of the defect was measured “(L 52.)?
Postoperative follow-up:
- Delay of initial hospital discharge?
- Mean follow-up duration at last review of these seven patients?
Discussion section
This paragraph would be more relevant if better structured with under paragraphs (resume of results, originality of the technique compared to the literature and perspectives).
The authors detailed in the discussion some others thoracoscopic approaches but the advantages of the author’s technique could be mor highlighted.
Moreover, the authors added another procedure associated the suture technique represented by the incision of the edges of the diaphragm that seemed to play a relevant role in the efficacity of the technique.
This proposed novel thoracoscopic approach is interesting, but the authors might nuance the conclusion concerning:
- the indication of the technique for larger defects ? : the 7 cases presented in the paper seemed present good prognostic form of CDH with small defect (postnatal presentation, tolerance of thoracoscopy procedure) .
- the quality of life of the patients due to the short follow-up of these patients and taking into account that the selected patients usually have a good quality of life even with other surgical techniques.
Other : Fig 1 is not useful for practitioners who know something about CDH
Author Response
The authors described in this paper a new technique with thoracoscopic approach for the congenital diaphragmatic repair. This minimal invasive technique is well described and well-illustrated. The relevance and the interest of the paper would increase if the authors specified the context of patients that would benefit from this technique.
Introduction section:
This paragraph should be improved. The authors have to precise the aim of the paper: to develop new approach of closing the diaphragmatic defect without patch that could be used for large defects or to describe a new different thoracoscopic technique for the diaphragmatic repair compared to the previous published thoracoscopic approaches.
The chapter “introduction” is rewritten. We elucidate why a direct suture is preferable to a prosthetic patch and show it with a previous publication from Putnam et al. Our description of a finally “t-shaped” configuration of the seams is not mentioned before.
The new approach of the authors relies on previous thoracoscopic technique (Boguzs et al.) or others one that could be more detailed in this paragraph to understand why the authors decided to “change” the way to setting the seams.
The combination of a t-shape and a pericostal guidance of the seams is our unique way for CDH-repair. Neither in the publication of Bogusz et al. nor in others it is mentioned like this. The t-shape configuration of the seams provides mechanically the lowest tension to the closed diaphragmatic muscle and minimizes the likelihood for recurrences. The pericostal setting is mechanical the best anchorage. See also the chapters “intraoperative situation” and “discussion”.
Patients and methods section:
Patients and methods section should be rewritten, especially the methods of inclusion of patients for the choice of the surgical technique, details of the data collected for each patient.
Patients’ data (Lines 38 to 43) should be included in result section, these data represent descriptive results.
The chapter results is rewritten, shared into “patient data”, “intraoperative findings” and “postoperative follow up”.
L38. Were the seven patients successive patients on the study period? How many CDH patients are being treated each year in their hospital and with what surgical technique ?
The patients were such with a left sided CDH and stable conditions.
Each year we have 2-3 CDH’s in our hospital.
L39. What were the circumstances and delay of the diagnosis of CDH of the patients without prenatal diagnosis? Were these patients intubated before the surgical repair or only for surgical repair?
All patients were intubated, the mothers of the three without prenatal diagnosis of CDH missed or avoided their prenatal ultrasound. This happens sometimes, even in central Europe.
L40. It would be interesting to know the antenatal characteristics for antenatally diagnosed patients (LHR o/e; pulmonary volume o/e; liver position..)
LHR is now included in chapter 3 “patient data”.
L41. Could the authors explain the delivery by cesarian section for the patients with prenatal diagnosis? Were these patients intubated at birth or only for the surgical repair? What are the medical conditions to perform thoracoscopic repair (including ventilation and pulmonary hypertension) ? Is thoracoscopic repair feasible as a bedside surgery in their center ?
Once a CDH is diagnosed prenatally intubation after delivery is obligatorily in our clinic. Thoracoscopy is performed, if the patient is in a cardiorespiratory stable condition, even if the CDH is on the right side. We don’t perform TA in a bedside setting.
Methods
Intraoperative situation: Why did the authors use 3.0 Ethibond and not 2.0 Ethibond ?
The only reason is, that 3.0 Ethibond seems to be strong enough.
Results section:
This very short section should be completed according to these following points:
Intraoperative findings:
- mean operative duration?
It is now mentioned: 158,8 min. (range 104-295 min.).
- size of the defect? The authors mentioned in methods section that “the size of the defect was measured “(L 52.)?
Concerned to the Congenital Diaphragmatic Hernia Study Group (CDHSG) in this study we repaired defects of the size “B”. See chapter 4 “discussion”. It is explained after “What is a small defect of CDH, what a large one and where is the limit for primary closure?”
Postoperative follow-up:
- Delay of initial hospital discharge? It was 19 days (range 7-33 days).
- Mean follow-up duration at last review of these seven patients? The eldest patient of our study is 7 Years today, the youngest 1 year. From the age of two years, we see them one time a year up to aldulthood, if there are no clinical problems.
Discussion section
This paragraph would be more relevant if better structured with under paragraphs (resume of results, originality of the technique compared to the literature and perspectives).
It is completely rewritten with a clear structure.
The authors detailed in the discussion some others thoracoscopic approaches but the advantages of the author’s technique could be mor highlighted.
We explain it now more detailed.
Moreover, the authors added another procedure associated the suture technique represented by the incision of the edges of the diaphragm that seemed to play a relevant role in the efficacity of the technique.
This proposed novel thoracoscopic approach is interesting, but the authors might nuance the conclusion concerning:
- the indication of the technique for larger defects ? : the 7 cases presented in the paper seemed present good prognostic form of CDH with small defect (postnatal presentation, tolerance of thoracoscopy procedure) .
All patients had size “B” close to “C”,
- the quality of life of the patients due to the short follow-up of these patients and taking into account that the selected patients usually have a good quality of life even with other surgical techniques.
We think we should operate a few hundred patients with this method, than we see, if it provides a better quality of life. Therefore our publication is a suggestion for an alternative surgical way and a possibility to exchange knowledge with collegues to develop the procedures of CDH rapairs together.
Other : Fig 1 is not useful for practitioners who know something about CDH
Fig. 1 has a new description.

Reviewer 2 Report
The article reports about a new technique of thoracoscopic closing of CDH. It is an interesting new suture technique which can reduce the number of recidives.
I have got some questions about the content of your article
The introduction is very short. Could you please provide a broader view on surgical aspects for the repair. What do you mean by “gentle surgical handling”?
To materials and methods: The mixture between patients and operation method is somehow difficult to read. Could you put your general description of the surgical technique into the method section and the patient details to results? (Also the sentences about the tension test (lines 78 and 79) from the result section could be placed in to the method section.)
Could you describe the positioning of the baby broader? Is it positioned cross-table? Is any padding used for positioning? Do you use relaxation during and after the surgery?
Results: There are some questions about the patients. Were those seven neonates all neonates born with CDH in your department during that time or did you select those patients for this kind or surgery? (And if so: how did you select?)
Could you give some more information about the surgery? How long did the surgery last? How big were the defects? How many sutures did you use? Were all surgeries performed by a single surgeon? You write, there were no complications regarding the cardiovascular system. Were there any other complications? What was the length of stay? How long was the follow-up of the patients? How is it performed?
Author Response
The article reports about a new technique of thoracoscopic closing of CDH. It is an interesting new suture technique which can reduce the number of recidives.
I have got some questions about the content of your article
The introduction is very short. Could you please provide a broader view on surgical aspects for the repair.
In general a CDH can be closed by an open access and by a thoracoscopic approach. In one hand, it depends on the size of the defect, wich is preoperatively barely to measure properly and if there are abdominal organs (liver, splen, etc.) in the thorax. On the other hand, mostly, it is decided by the surgeon which option is to be chosen. In our center the preferred method is the thoracoscopic approach. Once the view into the thorax is possible, the definitive decision for the way of the operational access can be made. Either it works by thoracoscopy or the thorax has to be opened.
In this publication I describe the official definition of the sizes of CDH based on the Congenital Diaphragmatic Hernia Study Group (CDHSG)of 2013.
What do you mean by “gentle surgical handling”?
This means a very carefully maneuver during the operation, including the regulation of the intrathoracic pressure by CO2 to avoid impairment of the cardiorespiratory situation and the touching and grasping of the tissue.
To materials and methods: The mixture between patients and operation method is somehow difficult to read. Could you put your general description of the surgical technique into the method section and the patient details to results? (Also the sentences about the tension test (lines 78 and 79) from the result section could be placed in to the method section.)
This chapter is now tidied up and more comprehensive.
Could you describe the positioning of the baby broader? Is it positioned cross-table? Is any padding used for positioning?
Thank You for this question! I describe the positioning of the baby much more detailed.
Do you use relaxation during and after the surgery?
Intraoperatively: “Under complete relaxation by anesthesia three 3-mm trocars are …”
After surgery there is relaxation, too, but only as deep and long, as the mechanical ventilation requires.
Results: There are some questions about the patients. Were those seven neonates all neonates born with CDH in your department during that time or did you select those patients for this kind or surgery? (And if so: how did you select?)
It is now described in the chapter “discussion” from line 157 to 164: Between 2016 and 2021 we had four additional patients with CDH. They could not be included in this study because it was not possible to perform the technique: One patient had a muscular defect on the right side with liver elevation. The defect was closed by TA without pericostal sutures. The other three patients were closed by OA: Two patients had a complete left sided CDH and needed a large patch. One patient, also left sided, had an additional malrotation of the intestine with ectopic pancreatic tissue.
Could you give some more information about the surgery?
Now in chapter 2 “Intraoperative situation”, line 90 to 114
How long did the surgery last?
To read now in chapter 3 “Results” an “Intraoperative findinfs”: The average duration time for thoracoscopic repair of the CDH was 158,8 min. (range 104-295 min.).
How big were the defects? Almost 50% of the hemilateral area of the diaphragmatic muscle. Line 134 to 135.
How many sutures did you use?
One for the Central Point of Suture (CPS) and up to seven additional sutures in pericostal position. Line 108
Were all surgeries performed by a single surgeon?
Yes. Line 116
You write, there were no complications regarding the cardiovascular system.
Were there any other complications?
Fortunately none. Albeit the fluctuations of blood pressure, Oxygenation and urine production during the first days on ICU. Explicit no recurrence CDH.
What was the length of stay? How long was the follow-up of the patients?
How is it performed?
It is described now in chapter 3, “Postoperative follow up”. Line 144 to 152

Reviewer 3 Report
Introduction:
REV: Please explain the abbreviation TA when first mentioned in the text (apart from the abstract).
REV: You state, that you modified a technique published in 2020 beginning with 2016? – please explain in detail.
REV: “Additional we incise the edges of the diaphragmatic defect as a iatrogenic tissue sore for a reliable durability of the repaired region” Do you mean tissue source?
REV: “This shows a recurrence-free outcome up to 35 today in a series of seven newborns.” This does not belong to the introduction but to the results and/or discussion section.
REV: I miss aims and hypothesis.
Methods:
AUT: “Seven neonates, six male, one female, with left sided CDH born between 2016 and 2021.”
REV: You refer to a left sided Bochdalek hernia? Were the patients consecutive? Do you perform TA for all CDH patients? How did you (if you did) select these 7 for TA.
AUT: “Body weight had a mean 3190 (+/- 1010) g.”
REV: Either “The mean body weight was...” or “The patients had a mean body weight of…”
AUT: “Time from birth to operation was 19 42 (+/- 11,5) hours.”
REV: Use “.” For comma in English not “,” please.
REV: What was their pre-OP history (iNO, HFO, ventilation days, lung-head ratio, best oxygenation index…). How was their adaptation after birth (spontaneous breathing, initial intubation,…)?
REV: you switch between present and past tense passive form (is done and was done) throughout the manuscript. Please choose either.
REV: You present patient data. Did you obtain approval of your ethics committee?
REV: How do you prevent injuries of the subcostal vessels/intercostal nerves with this technique. Were there any cases of intercostal neuropathy post-OP (are the patients sufficiently old to test this now?)?
Results:
AUT: “Intraoperative findings: All Patients..”
REV: patients.
AUT: The 75 reposition of the herniated viscera showed no complications…”
REV: “…reduction…” not “reposition” “
AUT: “The outcome of all seven patients to 84 date is excellent, we had no recurrent hernias.”
REV: Please include information about the follow-up interval (mean, standard deviation and range) for those patients.
Discussion:
AUT: “A pericostal suturing guided by a Tuohy needle is described in a case report7 and an adapted PIRS method…”
REV: Please explain the abbreviation PIRS
AUT: “We perform a direct percutaneous intro-93 duction of the suture by its origin needle.”
REV: I do not understand the meaning of this sentence. Original needle, native needle…?
AUT: “We cut the surface by a scalpel”
REV: “…with…”
AUT: “As we summed up our impression about these cases we are able to give a reliable statement that this procedure performed by TA provides a normal quality of live and a recurrence free repair from 2016 up to today.”
REV: This is very nice, and I am confident you are right. However, this statement is not very scientific. If you make statements about the quality of life you have to assess it (QOL questionnaires). Usually, infants the QOL of parents is assessed. If you do so you have to use age and sex matched healthy controls. As mentioned above additional information is required on your follow-up interval – best would be a detailed table regarding the data (GA, birth weight, respiration, other support like iNO, catecholamines, age at operation, operation time, respirator days post-OP, ICU duration and interval between operation and last follow-up for all patients.
Figure 1:
REV: I would prefer an x-ray with a little more contrast. It would be nice to have an image without “ufnahme” in it
Figure 2:
REV: the illustration needs some explanation to it. Where are the ribs, where is the muscle, what are the lines…? Figure heading 2b – you write: “intraoperativ” this should be either “intraoperative” or “intra-operative” The situs needs some explanation too. Use “*” or other symbols to describe the different anatomical structures.
Figure 3:
AUT (heading): “The geometry of the seams inside the thorax and the prove of tension from outside”
REV: “…proof of tension…” for the illustration and situs (detailed description please see my comment above).
Figure 4:
AUT (heading): “The yarn run around the rib, preferably located between the rib and the subcostal vessels. Further 146 the cut in the pleuroperitoneal channel and the ideal insertion of the diaphragm at the rib.”
REV: “..runs..” additionally “yarn” is a very trivial term usually referring to a string used for stitching. Surgical suture or suture would be used preferably here. (The same is the case for “seams” in the heading of figure 3. Probably suture line or suture alone would be better).
Figure 6:
REV: I do not understand the course of the central venous line. Please explain your rationale to place a chest tube post-OP.
Figure 7:
AUT: “Radiograph of the same patient after 6 month”
REV: “…months”
Author Response
Introduction:
REV: Please explain the abbreviation TA when first mentioned in the text (apart from the abstract).
It is done. Line 15, see the new version: CDH_Pericostal_Suture_with_TA_Michel6
REV: You state, that you modified a technique published in 2020 beginning with 2016? – please explain in detail.
Correction is done. In the abstract and in chapter 1 “Introduction”, line 63 to 71.
REV: “Additional we incise the edges of the diaphragmatic defect as a iatrogenic tissue sore for a reliable durability of the repaired region” Do you mean tissue source?
I changed the word, now it is “iatrogenic wound”. Line 94
REV: “This shows a recurrence-free outcome up to 35 today in a series of seven newborns.” This does not belong to the introduction but to the results and/or discussion section.
Now it is removed from the introduction to “discussion”. Line 227
REV: I miss aims and hypothesis.
You can read it now in the introduction. Line 69 to71
Methods:
AUT: “Seven neonates, six male, one female, with left sided CDH born between 2016 and 2021.”
REV: You refer to a left sided Bochdalek hernia? Were the patients consecutive? Do you perform TA for all CDH patients? How did you (if you did) select these 7 for TA.
It is rewritten in more details in chapter 3 “Results”. Line 119-133. And in chapter 4 “Discussion”. Line 154 to 164.
AUT: “Body weight had a mean 3190 (+/- 1010) g.”
REV: Either “The mean body weight was...” or “The patients had a mean body weight of…”
Thank You for this correction. See Line 132
AUT: “Time from birth to operation was 19 42 (+/- 11,5) hours.”
REV: Use “.” For comma in English not “,” please.
Thank You for this correction. See Line 132 to 133
REV: What was their pre-OP history (iNO, HFO, ventilation days, lung-head ratio, best oxygenation index…). How was their adaptation after birth (spontaneous breathing, initial intubation,…)?
I only found data to lung-head ratio, Line 126 to 127. Unfortunately none to oxygenation indices. A HFO was not used in any patient. Ventilation days were 75,8 hours (range 25,5-132 hours). Line 146
REV: you switch between present and past tense passive form (is done and was done) throughout the manuscript. Please choose either.
Thank You. I think it is corrected now.
REV: You present patient data. Did you obtain approval of your ethics committee?
Our ethics committee give us the permission for this publication.
REV: How do you prevent injuries of the subcostal vessels/intercostal nerves with this technique. Were there any cases of intercostal neuropathy post-OP (are the patients sufficiently old to test this now?)?
To prevent injuries, the semicircular needle helps to lead the seam in steady contact along the lower surface of the rib.
In the follow up we had baby or kid wich is suspected to have pain in the thoracic wall. All wounds have been healed properly.
Results:
AUT: “Intraoperative findings: All Patients..”
REV: patients.
Thank You.
AUT: The 75 reposition of the herniated viscera showed no complications…”
REV: “…reduction…” not “reposition” “
Thank You.
AUT: “The outcome of all seven patients to 84 date is excellent, we had no recurrent hernias.”
REV: Please include information about the follow-up interval (mean, standard deviation and range) for those patients.
Now it is mentioned in chapter 3 “Results”, “Postoperative follow up”. Line 147 to 152.
Discussion:
AUT: “A pericostal suturing guided by a Tuohy needle is described in a case report7 and an adapted PIRS method…”
REV: Please explain the abbreviation PIRS
This is now written in Line 207 and in the abbreviation list.
AUT: “We perform a direct percutaneous intro-93 duction of the suture by its origin needle.”
REV: I do not understand the meaning of this sentence. Original needle, native needle…?
Yes, “original”. Line 213.
AUT: “We cut the surface by a scalpel”
REV: “…with…”
Yes, “with”. Line 218.
AUT: “As we summed up our impression about these cases we are able to give a reliable statement that this procedure performed by TA provides a normal quality of live and a recurrence free repair from 2016 up to today.”
REV: This is very nice, and I am confident you are right. However, this statement is not very scientific. If you make statements about the quality of life you have to assess it (QOL questionnaires). Usually, infants the QOL of parents is assessed. If you do so you have to use age and sex matched healthy controls. As mentioned above additional information is required on your follow-up interval – best would be a detailed table regarding the data (GA, birth weight, respiration, other support like iNO, catecholamines, age at operation, operation time, respirator days post-OP, ICU duration and interval between operation and last follow-up for all patients.
Thank You for this comment. I agree, that this data would increase the scientific value. I suggest that it would make sense in larger series of patients with CDH. In this publication the focus is on the surgical technique. For instance, a data gathering about qualitiy of life after repair of CDH in Austria and Germany will be possible in near future by the “Kinderregister für angeborene Fehlbildungen, KiRaFe“, www.kirafe.org.
Figure 1:
REV: I would prefer an x-ray with a little more contrast. It would be nice to have an image without “ufnahme” in it
Thank You, I improved it.
Figure 2:
REV: the illustration needs some explanation to it. Where are the ribs, where is the muscle, what are the lines…?
I put a * into the area of the intrathoracic bowel.
Figure heading 2b – you write: “intraoperativ” this should be either “intraoperative” or “intra-operative”
Thank You, I improved it.
The situs needs some explanation too. Use “*” or other symbols to describe the different anatomical structures.
Thank You, I improved it.
Figure 3:
AUT (heading): “The geometry of the seams inside the thorax and the prove of tension from outside”
REV: “…proof of tension…” for the illustration and situs (detailed description please see my comment above).
Thank You, I improved it.
Figure 4:
AUT (heading): “The yarn run around the rib, preferably located between the rib and the subcostal vessels. Further 146 the cut in the pleuroperitoneal channel and the ideal insertion of the diaphragm at the rib.”
REV: “..runs..” additionally “yarn” is a very trivial term usually referring to a string used for stitching. Surgical suture or suture would be used preferably here. (The same is the case for “seams” in the heading of figure 3. Probably suture line or suture alone would be better).
Yes, good, thank You, I improved it.
Figure 6:
REV: I do not understand the course of the central venous line. Please explain your rationale to place a chest tube post-OP.
We use it regularely to control possible leakage of the lung and to support its unfolding.
The pleural drainage is explained in chapter 3 “Methods”. “Intraoperative Situation”. Line 113
Figure 7:
AUT: “Radiograph of the same patient after 6 month”
REV: “…months”
Thank You, I improved it.

Round 2
Reviewer 2 Report
After the revision the article improved much. There are only some minor tasks, which could further improve the article.
Line 104: is the intubation after 2-4 hours general treatment, or results of the retrospective chart analysis? If this is general treatment, what is the indication in a pulmonary stable child?
Line 164: “A retrospective analysis was performed 164 reviewing the medical data of seven neonates” this belongs to methods.
Line 195: Could you please add the mean follow-up time of your patients?
Lines 210-220 belong to results, not to discussion as they present new information on the cohort.
Author Response
"Line 104: is the intubation after 2-4 hours general treatment, or results of the retrospective chart analysis? If this is general treatment, what is the indication in a pulmonary stable child?"
We do this in general to prevent abdominal distension and to avoid an additional compromise of the lung function. It is now written in line 77-79.
"Line 164: “A retrospective analysis was performed reviewing the medical data of seven neonates” this belongs to methods."
Now You find it in Chapter 2. "Method", line 74-75
"Line 195: Could you please add the mean follow-up time of your patients?"
Up to today the mean follow-up time is 3.5 (1.1-6.3) years but the number will grow. Each patient was seen on average 5 (3-9) times after discharge and they are still in our follow up schedule. See line 152-154
"Lines 210-220 belong to results, not to discussion as they present new information on the cohort."
To this comment I can not create a reference in the article. May be I have a different refer to the lines. Could You please include a citation?